# Molecular and Biophysical Perspectives on Dormancy Breaking: Lessons from Yeast Spore

**DOI:** 10.3390/biom15050701

**Published:** 2025-05-11

**Authors:** Keiichiro Sakai, Yohei Kondo, Kazuhiro Aoki, Yuhei Goto

**Affiliations:** 1Department of Biology, Brooklyn College, The City University of New York, 2900 Bedford Avenue, Brooklyn, NY 11210, USA; sakai.keiichiro@brooklyn.cuny.edu; 2Center for One Medicine Innovative Translational Research (COMIT), Nagoya University, 65 Tsurumai-cho, Showa-ku, Nagoya 466-8550, Japan; kondo.yohei.e8@f.mail.nagoya-u.ac.jp; 3Laboratory of Cell Cycle Regulation, Graduate School of Biostudies, Kyoto University, Yoshidakonoe-cho, Sakyo-ku, Kyoto 606-8501, Japan; aoki.kazuhiro.6v@kyoto-u.ac.jp; 4Center for Living Systems Information Science, Graduate School of Biostudies, Kyoto University, Yoshidakonoe-cho, Sakyo-ku, Kyoto 606-8501, Japan; 5Division of Quantitative Biology, National Institute for Basic Biology, National Institutes of Natural Sciences, 5-1 Higashiyama, Myodaiji-cho, Okazaki 444-8787, Aichi, Japan; 6Quantitative Biology Research Group, Exploratory Research Center on Life and Living Systems (ExCELLS), National Institutes of Natural Sciences, 5-1 Higashiyama, Myodaiji-cho, Okazaki 444-8787, Aichi, Japan

**Keywords:** dormancy, dormancy breaking, spore, germination, fission yeast, budding yeast, cytoplasmic fluidity

## Abstract

Dormancy is a physiological state that enables cells to survive under adverse conditions by halting their proliferation while retaining the capacity to resume growth when conditions become favorable. This remarkable transition between dormant and proliferative states occurs across a wide range of species, including bacteria, fungi, plants, and tardigrades. Among these organisms, yeast cells have emerged as powerful model systems for elucidating the molecular and biophysical principles governing dormancy and dormancy breaking. In this review, we provide a comprehensive summary of current knowledge on the molecular mechanisms underlying cellular dormancy, with particular focus on the two major model yeasts: *Saccharomyces cerevisiae* and *Schizosaccharomyces pombe*. Recent advances in multifaceted approaches—such as single-cell RNA-seq, proteomic analysis, and live-cell imaging—have revealed dynamic changes in gene expression, proteome composition, and viability. Furthermore, insights into the biophysical properties of the cytoplasm have offered new understanding of dormant cell regulation through changes in cytoplasmic fluidity. These properties contribute to both the remarkable stability of dormant cells and their capacity to exit dormancy upon environmental cues, deepening our understanding of fundamental cellular survival strategies across diverse species.

## 1. Introduction

Living organisms have evolved diverse strategies to survive under unfavorable environmental conditions. Among these strategies, dormancy represents a remarkable adaptation that enables organisms to temporarily suspend their growth and metabolism while maintaining the capacity to resume proliferation when conditions become favorable [1,2]. This survival strategy is widely conserved across living organisms, from prokaryotes to eukaryotes. In bacteria, endospore formation, most notably studied in *Bacillus subtilis*, allows long-term survival under extreme conditions, including resistance to heat, radiation, and chemical exposure [3]. Among fungi, various species produce specialized dormant structures: *Aspergillus* and *Penicillium* form highly resistant conidia [4], while pathogenic fungi like *Cryptococcus* produce spores crucial for both survival and host infection [5]. In the plant kingdom, seeds represent a complex form of dormancy, allowing embryos to survive for years or even centuries while awaiting suitable germination conditions [6]. Even more remarkably, several animals have evolved sophisticated dormancy strategies. Tardigrades enter cryptobiosis (meaning ‘hidden life’), a term coined by David Keilin [7] and defined as ‘the state of an organism when it shows no visible signs of life and when its metabolic activity becomes hardly measurable, or comes reversibly to a standstill’ [8]. The cryptobiosis enables their survival in extreme environments, including space exposure [9]. Similarly, the larvae of the African midge *Polypedilum vanderplanki*, known as “sleeping chironomids,” can survive complete desiccation through anhydrobiosis (meaning ‘life without water’), a state of cryptobiosis that is induced by desiccation, maintaining their viable state for over one year through the accumulation of trehalose and heat shock proteins [10,11]. The ubiquity of dormancy across such diverse taxa underscores its fundamental importance as a survival strategy. Dormancy is characterized by several distinct features: (1) dramatic reduction in metabolic activity, (2) enhanced resistance to various environmental stresses, (3) reversible transition between dormant and proliferative states (dormancy breaking), and (4) long-term survival capability. These characteristics provide significant evolutionary advantages, allowing organisms to persist through periods of nutrient limitation, temperature extremes, or other environmental challenges. The ability to enter and exit dormancy in response to environmental cues is crucial for the survival and propagation of living organisms in fluctuating environments.

*Saccharomyces cerevisiae* and *Schizosaccharomyces pombe* have emerged as powerful models for studying cellular dormancy. These two yeast species diverged approximately 330–420 million years ago, with an evolutionary distance comparable to that between yeasts and mammals [12,13]. Accordingly, comparative studies of these phylogenetically distant yeasts are expected to provide insights into both the evolutionary conservation and diversification of cellular dormancy. Nitrogen starvation induces the formation of spores, a dormant state, in these yeasts when they exist as diploid cells [14]. During this sporulation process, diploid yeast cells undergo meiosis to form haploid spores, which are dormant cells equipped with specialized cell walls and distinct metabolic properties [15,16]. These spores remain viable for extended periods and break dormancy, i.e., germination, to resume vegetative growth when favorable conditions return. Beyond sporulation, yeasts are also able to enter quiescence during the stationary phase of vegetative growth [17,18]. Both dormancy and quiescence refer to reversible growth-arrested states, but dormancy is generally thought to be a “deeper” form of quiescence [19]. Quiescent cells typically retain higher levels of intracellular activity compared to dormant cells. Furthermore, they differ in their formation mechanisms, stress resistance properties, and metabolic characteristics [20]. This review primarily focuses on spore dormancy and germination.

Here, we discuss the molecular and biophysical mechanisms underlying yeast spore dormancy and germination, with particular emphasis on recent findings from *S. cerevisiae* and *S. pombe*. Table 1 provides a comparative overview of the key features of dormancy and germination in these two yeast species, highlighting their similarities and differences. We further explore emerging questions in this rapidly evolving field, with particular focus on molecular, biophysical, and ecological perspectives.

**Table 1 biomolecules-15-00701-t001:** Comparison of Dormancy and Germination between *S. cerevisiae* and *S. pombe*.

Feature	*S. cerevisiae* (Budding Yeast)	Refs	*S. pombe* (Fission Yeast)	Refs
Life Cycle Characteristics				
Natural state	Predominantly diploid	[21]	Predominantly haploid	
Mating type	MATa and MATα		h- and h+	
Sporulation Conditions				
Primary trigger	Nitrogen depletion with non-fermentable carbon source	[22]	Nitrogen depletion	
Spore Structure				
Ascus wall after sporulation	Remains intact		Digested by glucanases (Agn2, Eng2)	[23,24,25]
Spore wall composition	Four layers: mannan, β-1,3-glucan, chitosan, dityrosine	[15,26,27]	Primarily glucan and chitosan (exact composition not fully characterized)	[28,29,30]
Spore connection	Interspore bridges connect sibling spores	[31]	No interspore bridges	
Spore surface	Ridged proteinaceous layer	[32]	Characteristic outward projection	[33,34]
Metabolic Features				
Trehalose accumulation	Significant increase	[35,36]	~1000-fold increase compared to vegetative cells	[37,38,39]
Glycogen accumulation	Present	[35,36]	~40-fold increase compared to vegetative cells	[37,39]
Adenosine triphosphate (ATP) levels	Substantial (~3–4 mM)	[35]	Relatively high compared to residual ascus	[40]
Transcription activity	~5% of vegetative cells	[41]	Not precisely quantified	
Protein filament formation	Acetyl-CoA synthetase Acs1 forms filaments	[42]	Not known	
Germination Process				
Typical duration	4–6 h	[43,44,45,46]	10–12 h	[37,47,48,49]
Primary trigger	Glucose (Cyclic adenosine monophosphate-protein kinase A (cAMP-PKA) pathway)	[50]	Glucose (cAMP-PKA pathway)	[37,47,51]
Initial stages	Spore uncoating, polarized growth	[43,44,45,46]	Bright-to-dark transition, isotropic swelling	[37,47,48]
Polarization mechanism	Prepolarized to grow away from interspore bridges	[52]	Random polarization with dynamic polar cap movement	[48]
Polarization proteins	Cdc10, Bud8, Bud5	[52,53]	GTP-bound Cdc42, Bud6, Bgs4, Cdc42 GAP (Rga6)	[48,54,55]
Cell growth pattern	Polarized growth -> Non-polarized growth -> Budding	[43,44,45,46]	Isotropic swelling -> Germ tube formation (outgrowth)	[37,47,48,49]
Required Nutrients for Complete Germination				
Glucose	Required for fast response	[44,50,56,57]	Essential	[58]
Additional nutrients	Required for later stages	[44,50,57]	Copper and iron ions required for outgrowth	[59,60]
Molecular Regulators				
Key signaling pathway	cAMP-PKA pathway	[50]	cAMP-PKA pathway	[37,47,51]
Trehalase	Nth1, Nth2		Ntp1	[37,38]
Cell cycle regulators	Not required for early germination stage (Cdc28, Cdc37, Cdc4, Cdc34, Cdc7, Cdc24)	[50]	Not known	
Actin role	Essential for polarized growth	[46]	Essential for germ tube formation	[47]
Histone dynamics	Not known		Hht1 (H3) expression decreases during germination	[61]
Heat shock protein	Hsp42	[62]	Not known	
Biophysical Properties				
Particle mobility in dormant spores	Restricted (~50–150 nm particles)	[62]	Restricted (~40 nm and ~50–150 nm particles)	[37]
Small protein diffusion	Not fully documented		Relatively free diffusion	[37]
pH changes during germination	Dormant spores: ~5.9 -> Vegetative cells: ~7.4	[62,63]	Not known	
Ecological Context				
Natural habitat	Fruits, insect vectors, forest niches	[64,65,66]	Not well characterized (honey?)	[67]
Spore survival advantage	High survival in the insect gut	[64]	High survival in the insect gut	[64]
Germination pattern	Commonly sibling spore mating	[52,68]	Single spores	

## 2. Dormancy in Yeast Cells

### 2.1. S. pombe

The life cycle of the fission yeast *S. pombe* is shown in Figure 1A. *S. pombe* naturally exists as haploid cells with two mating types; minus (h−) and plus (h+). In response to nitrogen depletion, *S. pombe* cells arrest in the G1 phase, and heterothallic haploid cells of opposite mating types conjugate to form diploid zygotes. In contrast, when haploid cells fail to encounter the opposite mating type, they enter a quiescent state. These diploid zygotes undergo sporulation, following chromosome recombination and meiosis. Transcriptome, proteome, and phosphoproteome profiles drastically change throughout meiosis and sporulation [69,70,71]. A genome-wide screening using transposon insertion sequencing identified 532 genes contributing to meiosis and sporulation, and a screening using a non-essential gene deletion library identified 34 genes contributing to them [72,73].

#### 2.1.1. Enhanced Stress Resistance of *S. pombe* Spores

Sporulation yields four identical spores with a characteristic outward projection as a surface structure [33,34]. These spores possess a spore wall, a structure thicker than the cell wall of vegetative cells. Although its composition remains unknown in *S. pombe* spores, it is thought to be primarily composed of glucan and chitosan [28,29,30]. *S. pombe* spores exhibit high resistance to various environmental stresses, such as heat, organic solvents (e.g., ethanol), digestive enzymes (e.g., glusulase), and nutrient starvation [16,58,74]. They can survive for several months even in an aqueous solution with almost no nutrients, demonstrating higher survival rates compared to cells in the stationary phase and quiescence [16]. The spore coat protein Isp3, localized to the outermost layer of the spore wall, appears to confer this remarkable stress tolerance [74]. This enhanced stress resistance provides a survival advantage in natural environments; for example, spores exhibit significantly higher viability than vegetative cells when present inside the body of flies [64].

#### 2.1.2. Metabolic Changes in *S. pombe* Spores

Although global metabolic changes during *S. pombe* sporulation remain uncharacterized, specific metabolites show dramatic alteration in spores; trehalose and glycogen levels increase 1000-fold and 40-fold, respectively, compared to vegetative cells [37,38,39]. These changes in carbohydrate levels are proposed to regulate cytoplasmic fluidity in dormant spores [37] (refer to Section 4.2. Dormancy and Dormancy Breaking). Spore cytoplasm maintains relatively high ATP levels compared to the residual ascus [40]. Furthermore, lipid droplets, primarily composed of triacylglycerols and sterol esters, are actively redistributed from mother cells to their nascent spores. This redistribution depends on actin polymerization and is essential for proper spore germination, as the absence of lipid droplets results in germination defects [75].

### 2.2. S. cerevisiae

The life cycle of the budding yeast *S. cerevisiae* is shown in Figure 1B. In the natural environment, the budding yeast *S. cerevisiae* predominantly exists as diploid cells [21] with two mating types; *MATa* and *MATα*. These a/α diploid cells undergo meiosis and sporulation when exposed to nitrogen-depleted conditions in the presence of a nonfermentable carbon source, such as acetate, ethanol, and pyruvate [22]. By forming dormant spores, yeast cells acquire resistance to a variety of environmental stresses, enhancing their survival in harsh conditions [15]. Notably, sporulation functions as a rejuvenation by eliminating age-associated cellular damage, including protein aggregation and nucleolar aberrations [76]. Consequently, spores derived from aged cells exhibit replicative potential equivalent to those from young cells [76].

#### 2.2.1. Enhanced Stress Resistance of *S. cerevisiae* Spores

The budding yeast spores exhibit remarkable resistance to various stresses, including heat, desiccation, organic solvents, and digestive enzymes [15,77,78]. This stress resistance is largely mediated through the spore wall [15,79,80,81]. The spore wall, which is thicker than the vegetative cell wall, consists of four distinct layers; mannan, β-1,3-glucan, chitosan, and dityrosine layers. The two inner layers are composed of mannan and β-1,3-glucan, and the third and fourth layers are composed of chitosan and dityrosine, respectively [15,26,27]. While mannan and β-1,3-glucan layers are present in both spore and vegetative cell walls, the chitosan and dityrosine layers are unique to the spore wall [15,79,81]. Mutation of DIT1, which encodes a dityrosine precursor enzyme, specifically disrupts the dityrosine layer without affecting the chitosan layer [79]. The mutant of the chitin synthase CHS3 leads to the loss of both chitosan and dityrosine layers while preserving the inner layers of the spore wall [81]. Spores lacking either the dityrosine layer alone or both the chitosan and dityrosine layers show increased stress sensitivity, indicating the critical role of these layers in stress resistance [79,81]. The dityrosine layer appears to function as a diffusion barrier, preventing the entry of protein-sized molecules into the spore wall, thereby potentially explaining its role in conferring resistance to digestive enzymes [15,80]. A recent study reported that *S. cerevisiae* spores possess an additional surface layer above the dityrosine layer [32]. This ridged proteinaceous layer is suggested to function as a protective structure.

#### 2.2.2. Reduced Metabolic Activities in *S. cerevisiae* Spores

Dormant spores are thought to be metabolically inactive. Transcription and translation activities in spores at 30 °C incubation are approximately 5% of those observed in vegetative cells [41]. In addition, mRNAs remain stable for several months in spores stored at 4 °C, suggesting that both synthesis and degradation of cellular components are markedly suppressed during dormancy. Using a doxycycline-inducible synthetic gene-circuit, it was demonstrated that GFP expression reaches a plateau after approximately 20 h in spores, compared to 8 h in vegetative cells [82]. This observation suggests a slowdown in gene expression in spores, although the delay may also reflect the unique physiological environment within spores that affects oxygen-dependent GFP chromophore maturation. This reduced metabolic activity likely promotes spore survival under harsh environmental conditions, particularly during nutrient starvation where energy generation and biosynthesis are limited. Notably, pharmacological inhibition of either ATP synthesis or translation for approximately 24 h does not compromise spore viability [82]. In contrast, transcriptional inhibition results in the death of almost all spores, suggesting that minimal transcriptional activity is necessary for spore survival [82]. Since transcription initiation requires ATP as an energy source [83], it is speculated that dormant spores retain sufficient ATP to sustain minimal transcription for at least 24 h in the absence of ATP synthesis. Another study identified the assembly of metabolic enzymes into filaments within spores by combining cryo-electron tomography (cryoET) and single-particle cryo-electron microscopy (cryoEM) [42]. Specifically, acetyl-coenzyme A (CoA) synthetase Acs1 forms filaments that suppress acetyl-CoA production in the spore [42]. This filament-mediated inactivation of Acs1, a major ATP-consuming enzyme, may help maintain ATP levels necessary for minimal metabolic activity [42]. Indeed, metabolomic analyses have revealed that spores maintain substantial amounts of ATP (~3–4 mM) [35], comparable to those found in vegetative cells [40,84].

#### 2.2.3. Carbohydrate Accumulation in *S. cerevisiae* Spores

Carbon storage represents a conserved feature of cells in dormant and quiescent states [2]. Similar to *S. pombe* spores, *S. cerevisiae* spores accumulate specific carbohydrates, particularly trehalose and glycogen. During meiosis and sporulation in budding yeast, trehalose synthesis is actively induced [35,36]. This process is catalyzed by Tps1, a subunit of the trehalose-6-phosphate synthase/phosphatase complex, which serves as the key enzyme for trehalose biosynthesis [85]. Diploid cells lacking Tps1 exhibit significantly reduced sporulation efficiency compared to wild-type cells [86], indicating that trehalose accumulation is essential for faithful sporulation. While glycogen also accumulates in mature spores [35,36], it undergoes transient degradation following key meiotic events, potentially providing precursors for spore wall assembly [35]. Similar patterns of trehalose and glycogen accumulation occur in quiescent and G1-arrested states in *S. cerevisiae* cells [87,88,89], where these carbohydrates likely provide energy to fuel cell cycle re-entry upon growth resumption. By analogy, stored carbohydrates in spores may serve as a crucial energy source during germination progression after release from dormancy.

#### 2.2.4. Ecological Significance of *S. cerevisiae* Spores

The ecological significance of sporulation in budding yeast has been revealed through various studies. *S. cerevisiae* has been isolated from a variety of natural habitats, especially from fruits, while also establishing associations with insect vectors, such as social wasps, flies, and other insects [64,65,66]. In insect vectors such as the fruitfly *Drosophila melanogaster*, spores exhibit remarkable resistance to digestive processes in their gut [64]. The paper demonstrates that spores pass through the fly gut with significantly higher survival rates compared to vegetative cells. Notably, when tetrads (spores enclosed within the ascus) pass through the fly gut, the ascus walls are digested, causing the dissemination of each spore [90]. This dissemination of spores enhances outbreeding rates more than 10-fold relative to intact tetrads, which increases genetic recombination between different yeast strains and promotes genetic diversity in the resulting progeny [90]. Beyond insects, *S. cerevisiae* has been isolated from forest niches, such as soil and tree bark. Although direct evidence for spores in these forest niches remains elusive, experimental studies have demonstrated that soil environments promote sporulation in *S. cerevisiae* [65]. These observations led to the development of the ‘fruit forest-reservoir hypothesis’, which proposes a tripartite ecological cycle. According to this model, *S. cerevisiae* primarily inhabits three distinct niches: fruits, insect vectors, and forest environments. During fruit season, yeast cells proliferate through mitotic division and fermentation within fruits. As fruits become scarce, some fraction of the cells migrate to forest environments via insect vectors or fallen fruits. In forest niches, cells adopt a sporulated state, enabling survival through nutrient-poor winter conditions.

## 3. Dormancy Breaking in Yeast Cells

### 3.1. S. pombe

Upon exposure to favorable conditions, *S. pombe* spores exit dormancy and initiate cell cycle progression and vegetative growth. Germination onset can be monitored using phase contrast microscopy [37,47,48]. Dormant spores appear as highly refractile, bright cells under phase contrast microscopic observation. When transferred to a nutrient-rich medium, spores begin germination, marked by a characteristic bright-to-dark transition of the cytoplasm 1–2 h after induction—this represents the earliest detectable hallmark of germination onset (germination sensu stricto) [37,47,48]. The germination process continues with isotropic swelling at 4 h, followed by germ tube elongation (outgrowth) at 6 h post-induction [37,47,48,49]. Subsequently, the first nuclear division occurs at 10 h with cytokinesis (septation) following at 12 h post-induction [37,47,48,49]. The entire sequence constitutes germination sensu lato [49]. The major events during spore germination are illustrated in Figure 2.

#### 3.1.1. Glucose-Sensing and Trehalose Degradation Pathway During Germination in *S. pombe*

Spore germination is typically triggered by a glucose-containing medium [47,58,91]. In *S. pombe*, glucose serves dual functions; it acts as both a carbon source supporting cell cycle progression and an activator of the glucose-sensing cAMP-PKA pathway [37,47,92]. Glucose recognition occurs through the G-protein coupled receptor, Git3, at the cell membrane, which triggers activation of the adenylate cyclase, Cyr1, via heterotrimeric G proteins, Gpa2 (Gα), Git5 (Gβ), and Git11 (Gγ). Activated Cyr1 produces cAMP as a second messenger, which binds to the PKA regulatory subunit Cgs1 and leads to the activation of the PKA catalytic subunit Pka1 [51,93,94]. Genetic analysis of spores lacking *git3*, *gpa2*, *cyr1*, or *pka1* genes revealed defective germination, demonstrating the essential role of the cAMP-PKA pathway in germination initiation [37,47,51]. PKA activation also promotes trehalose degradation during spore germination; wild-type spores degrade trehalose within one hour of germination induction, whereas *pka1*Δ spores maintain constant trehalose levels [37,38,39]. Trehalose degradation requires natural trehalase Ntp1, which hydrolyzes trehalose into two molecules of glucose [95]. Spores lacking Ntp1 fail to degrade trehalose upon germination induction, phenocopying *pka1*Δ [37,38]. Furthermore, the *ntp1*Δ spores also exhibit impaired germination initiation [37,38]. The germination defects in *gpa2*Δ spores are partially rescued by Ntp1 overexpression, indicating that Ntp1 is one of the major downstream factors in the cAMP-PKA pathway during germination [37]. Thus, the glucose-activated cAMP-PKA-Ntp1 signaling axis plays a central role in spore germination progression in *S. pombe*. While trehalose degradation is crucial, trehalose synthesis also influences the spore germination process: deletion of the *tps1*, encoding trehalose-6-phosphate synthase, prevents germination initiation [96]. Given the importance of trehalose degradation for spore germination, trehalose synthesis likely functions primarily to ensure spore survival under adverse environments rather than directly promoting the germination process. In the future, it will be of particular interest to identify downstream factors of the cAMP-PKA pathway and to elucidate their specific roles during germination. In *S. pombe*, a comprehensive analysis of PKA substrates remains unavailable, even for vegetative growth conditions. Consequently, our understanding of molecular pathways activated downstream of glucose-induced PKA signaling is limited, with only a few identified targets such as Ntp1. A promising approach would involve integrated transcriptome, proteome, and phosphoproteome analysis of the *pka1*Δ strain in vegetative cells, as synchronizing germination in *S. pombe* spores presents significant technical challenges (see Section 3.1.2. Gene Expression Landscape during Germination in *S. pombe*). Such multi-omics would likely reveal the comprehensive molecular networks that orchestrate the initiation of germination.

#### 3.1.2. Gene Expression Landscape During Germination in *S. pombe*

Recent single-cell transcriptomic analysis has revealed the gene expression landscape during spore germination in *S. pombe* [61]. The single-cell transcriptome was necessary because synchronizing germination progression in *S. pombe* spores remains technically challenging, in contrast to *S. cerevisiae* spores [43]. In this article, the authors identified 167 genes exhibiting dynamic expression changes during the early stage of germination. The enrichment analysis revealed associations with mRNA metabolic process, lipid metabolic process, nucleotide-containing small molecule metabolic process, ascospore formation, and detoxification. Among these, *hht1*, encoding histone H3, emerged as a particularly intriguing candidate. *S. pombe* possesses three histone H3-encoding genes (*hht1*, *hht2*, and *hht3*) with identical amino acid sequences. Among these, only the *hht1* expression shows dynamic fluctuation upon germination, while the *hht2* and *hht3* expressions remain relatively constant. Deletion analysis reveals that both *hht1*Δ and *hht2*Δ spores exhibit defects in germination progression, while *hht3*Δ spores germinate normally. Hht2 appears to function as a housekeeping histone because *hht2*Δ vegetative cells exhibit temperature sensitivity and growth defects, while *hht1*Δ and *hht3*Δ cells grow normally. These findings establish Hht1 as a germination-specific important histone, while Hht2 serves essential functions in both germination and vegetative growth. The reduction in Hht1 amount during the onset of germination may promote the relaxation of chromatin structure, facilitating a global transcriptional activation. For a comprehensive overview of other germination-related candidate genes and their expression patterns, readers are directed to the review by Tsuyuzaki et al. [49]. Determining the precise functions of these candidate genes during germination would be a valuable future direction. In addition, the established single-cell transcriptome methodology offers powerful opportunities when applied to mutant strains for characterizing gene expression dynamics throughout germination. A promising candidate for such analysis is Pka1, as discussed in Section 3.1.1, which could illuminate the molecular mechanisms governing germination in *S. pombe*.

#### 3.1.3. Nutrients Required for Germination in *S. pombe*

The progression of entire germination requires the appropriate nutrients beyond glucose. A simple glucose solution lacking other basic nutrients can trigger initial germination, which is measured by the characteristic bright-to-dark transition of spore cytoplasm observed under phase contrast microscopy [58]. Similar initial germination responses can be triggered by simple sucrose and mannose solutions [58]. However, subsequent morphological changes, such as spore swelling and germ tube formation, fail to occur in these minimal solutions, indicating that additional nutrients are necessary for later stages of germination. Copper and iron ions play crucial roles in spore outgrowth [59,60]. Under copper- or iron-insufficient conditions, germinating spores arrest their differentiation immediately after isotropic swelling [59,60]. Successful outgrowth requires the uptake systems for the specific ions and biosynthesis in these conditions, including copper transporters Ctr4 and Ctr5, the copper sensor Cuf1, ferrichrome biosynthesis enzymes Sib1 and Sib2, and the transporter Str1 [59,60]. In the future, a systematic investigation of nutrient requirements for germination would yield valuable insights into the molecular mechanisms that drive germination progression. Currently, glucose function is well established; it acts both as a primary carbon source and an activator of the cAMP-PKA signaling pathway. In contrast, our knowledge of other nutrients, particularly copper and iron ions, remains limited, and many potentially essential nutrients await discovery and characterization. Deciphering the complete nutritional landscape necessary for germination would likely reveal important regulatory mechanisms governing this fundamental biological process.

#### 3.1.4. Morphological Changes During Germination in *S. pombe*

The molecular mechanisms underlying morphological changes during spore germination have been extensively investigated. Germinating spores undergo two distinct morphological changes: isotropic swelling and germ tube formation (outgrowth). Inhibition of either protein translation or target of rapamycin (TOR) signaling, which regulates ribosome biogenesis, prevents these morphological changes [37]. These observations demonstrate that protein synthesis is required for proper germination progression. Consistent with this, protein content increases linearly during spore outgrowth [91]. Cytoskeletal organization, involving both actin and microtubule networks, plays essential roles in germ tube formation [37,48]. F-actin patches initially distribute randomly throughout the cortical region of germinated spores and subsequently become concentrated at a specific cortical site in swollen spores. After that, germ tube formation occurs at this actin-enriched site [47]. This actin localization precedes germ tube formation, indicating that cell polarity is established during the spore swelling phase. A recent study has elucidated the molecular mechanisms governing this cell polarization during spore germination [48]. The vegetative fission yeast cells establish polarity through the formation of a polar cap at the growing tip. This polar cap consists of various polarity proteins, including a Rho-type GTPase Cdc42, actin regulators, and cell-wall remodeling proteins. Germinating spores establish a single polar cap that exhibits dynamic behavior, undergoing stochastic movement across the cell surface through cycles of assembly and disassembly. The polar cap eventually stabilizes at a specific site when spores reach approximately twice their initial volume, making the location of future germ tube emergence. The polar cap includes GTP-bound Cdc42, actin nucleation-promoting factor Bud6, and (1,3)β-D-glucan synthase Bgs4 [48,54]. The site where the polar cap is stabilized has a singular rupture in the outer spore wall (OSW), releasing OSW constraints on polarized cell growth. Therefore, successful outgrowth requires both polar cap stability and appropriate spore wall mechanics. The spatial regulation of this process is further controlled by Cdc42 GTPase-activating proteins (GAP), which is necessary for establishing monopolar outgrowth during germination [55]. Although three Cdc42 GAPs (Rga3, Rga4, and Rga6) have been identified in *S. pombe* [97], only Rga6 has a key role in maintaining GTP-bound Cdc42 at the outgrowth zone [55].

### 3.2. S. cerevisiae

The spore germination process of *S. cerevisiae* can be divided into roughly four stages from the perspective of morphological changes: (1) spore uncoating, (2) polarized cell growth, (3) non-polarized cell growth, and (4) bud emergence [43,44,45,46]. Following these morphological changes, yeast spores re-enter the mitotic cell cycle and restart vegetative growth. Upon germination induction with a nutrient-rich medium, the spore cytoplasm exhibits a decreased electron density with swelling, followed by local degradation of the outer two spore layers: dityrosine and chitosan (spore uncoating) [46,62]. The inner two spore layers (mannan and β-1,3-glucan) remain intact and subsequently function as the vegetative cell wall [98]. During the next phase, the spore elongates unidirectionally along its long axis without any obvious growth along its short axis (polarized cell growth). Subsequently, the spore expands uniformly in both the long and short axes until bud emergence (non-polarized cell growth). *S. cerevisiae* spores complete their entire germination process in approximately 4 to 6 h, which is relatively fast compared to the 10 to 12 h required for *S. pombe* spores. The major events during spore germination are illustrated in Figure 3.

#### 3.2.1. Characteristics for *S. cerevisiae* Germination: Intact Ascus and Interspore Bridge

One of the unique features of *S. cerevisiae* is that the ascus wall remains intact after sporulation is completed, whereas in *S. pombe*, the ascus wall is digested by glucanases, Agn2 and Eng2, immediately after the completion of meiosis [23,24,25]. In addition to the intact ascus wall, *S. cerevisiae* spores are interconnected by interspore bridges, which hold spores closely even after digestion of the ascus wall [31]. Unlike *S. pombe*, *S. cerevisiae* naturally exists as diploid cells. A potential role of the intact ascus wall and the interspore bridge is to keep sibling spores close, enabling efficient mating and producing diploids after germination begins. Indeed, germinating spores of *S. cerevisiae* can mate with spores of an opposite mating type before budding and the mitotic cell cycle [52,68]. Global gene expression analysis has revealed that genes associated with mating are upregulated in the early germination stage [43,44]. In naturally isolated strains, mating between sibling spores is the usual event that follows germination [68]. The mating process has been mainly investigated with haploid lab strains, which are heterothallic and incapable of self-mating; thus, further studies focused on homothallic wild yeasts would provide intriguing insights into mating behavior. Another unique feature of *S. cerevisiae* is that spores are inherently prepolarized to outgrow away from interspore bridges, which occurs before budding or mating [52]. As mentioned above, *S. pombe* spores undergo symmetry breaking and polarization at random sites on the cell surface during germination (refer to Section 3.1.4. Morphological Changes during Germination in *S. pombe*) [48]. In *S. cerevisiae* spores, prepolarized sites contain the septin Cdc10, the cortical landmark protein Bud8, and the GTPase Rsr1 GEF Bud5 [52,53].

#### 3.2.2. Methods to Monitor *S. cerevisiae* Germination

Germination progression can be monitored using various methods in *S. cerevisiae* spores. Numerous studies have determined the onset of germination by purifying spores before inducing germination and assessing the loss of stress resistance in the spores. For example, after germination onset, budding yeast spores are sensitive to heat shock and digestive enzymes (e.g., Zymolyase) [43,50,62,99]. The increased stress sensitivity can be attributed to spore uncoating, which leads to the rupture of stress-resistant spore wall structures. Additionally, optical density decreases concurrently with increased sensitivity or slightly later [56,62,100,101,102]. Similar to *S. pombe* spores, the spore cytoplasm exhibits a bright-to-dark transition under the phase contrast microscope around the same time [62,101]. A recent study investigated spore germination by monitoring the emergence of elongated spores from the ascus following its rupture [82]. This approach is unique to budding yeast, unlike fission yeast, in which the ascus ruptures immediately after meiosis completion. The ascus rupture is due to the polarized cell growth and/or bud emergence, and therefore, this is an indicator of the later germination stage compared to increased stress sensitivity.

#### 3.2.3. Molecular Mechanisms Regulating *S. cerevisiae* Germination

The molecular mechanisms regulating germination progression have been studied in *S. cerevisiae*. As in *S. pombe*, the cAMP-PKA glucose-sensing pathway is required for the germination initiation [50]. Ras2 (GTP-binding protein), Cdc25 (GEF for Ras proteins), and Cyr1 (adenylate cyclase) are central components of this pathway, and the absence of these genes results in germination defects. This indicates that the cAMP-PKA pathway has a crucial role in the early germination stage, including spore uncoating. In contrast, many key cell cycle regulators are dispensable for the early germination stage. The spores lacking Cdc28 (cyclin-dependent kinase) and Cdc37 (Hsp90 co-chaperone required to pass through the START) undergo spore uncoating and polarized cell growth after germination induction. Other cell cycle regulators, including Cdc4 and Cdc34 (G1/S transition), Cdc7 (S phase), and Cdc24 (GEF for Cdc42), are also not necessary for the early germination stage. Furthermore, actin distribution dynamically changes during the germination process [46]. Actin patches randomly distribute upon spore uncoating, then accumulate at the tip region to support polarized growth, and disperse again after a few hours, probably promoting the non-polarized cell growth. Then, they localize to the bud site upon bud emergence. The spores treated with an actin depolymerizer, latrunculin A, exhibit no polarized growth even after germination induction. The polarized site includes polarisome proteins (Spa2, Pea2, Bud6, and Bni1), and these proteins are considered to establish cell polarization during germination. A recent study based on single-cell level analysis has revealed that dormant spores gradually lose their gene expression ability at 30℃ over days to months [82]. With this decreased ability, the amount of RNA polymerase II (RNAPII) also decreases over time. Interestingly, spores with higher levels of RNAPII are more likely to germinate than those with lower levels of RNAPII after approximately a month of incubation at 30℃. These results show that gene expression ability associated with RNAPII levels determines the germination ability. Other recent findings have shown that many proteins assemble to form cytoplasmic foci in dormant spores, which disperse upon germination induction [62,103]. The loss of Hsp42, a small heat shock protein, disrupts the dissolution of these foci, implying its involvement in controlling protein assembly during germination.

#### 3.2.4. Transcriptome, Proteome, and Phosphoproteome Profiles During *S. cerevisiae* Germination

One advantage of studying germination in budding yeast is the potential to achieve synchronization of germination timing. This approach allows us to investigate global changes in transcriptome, proteome, and phosphoproteome during germination. Transcriptomic analysis revealed that the germination process can be divided into two different stages after glucose addition [43,44]. The first stage (from germination start to ~1.5–2 h) is the gene expression pattern similar to the general response of budding yeast cells to glucose; glucose addition to stationary phase cells or glucose-starved cells. For example, genes associated with protein synthesis (e.g., ribosome biogenesis, rRNA processing) are upregulated, while genes related to stress response and growth on non-optimal carbon sources (e.g., gluconeogenesis, TCA cycle) are downregulated. However, there are some exceptions during germination (e.g., cell cycle-related genes) compared to the general glucose response, indicating that these differences make the germination-specific mechanisms. This gene expression in the first stage can also be induced when spores are incubated in glucose alone, not nutrient-rich medium. The second stage (from 2 h to the entering of the first mitotic cell cycle) makes spores capable of sensing and responding to environmental conditions other than glucose. For example, mating-related genes are upregulated, thereby allowing spores to respond to mating pheromones. Other transcriptomic analyses identified a complex transcription factor network regulating gene expression during germination [104]. Additionally, recent proteome analysis identified the changes in global protein solubility during germination [62]. In particular, metabolic enzymes involved in carbohydrate, lipid, and nitrogen metabolisms solubilize during germination, reflecting the activation of spore metabolism. In the study, phosphoproteome analysis determined the dynamic phosphorylation of S233 of Hsp42 during germination. The Hsp42 phosphorylation is correlated with its solubility and is crucial for the metabolic enzyme Acc1 solubilization. Therefore, these dynamic changes in transcriptome, proteome, and phosphoproteome provide insight into the comprehensive mechanisms to regulate germination progression. Investigating the relationship between enzyme solubilization and subsequent enzymatic activity during the initiation of germination represents a compelling avenue for future research.

## 4. The Biophysical Properties of the Cytoplasm During Dormancy

### 4.1. Overview

The cytoplasm is a highly complex and crowded milieu, densely packed with diverse cellular components, such as proteins, nucleic acids, and organelles. The crowded conditions impact the biochemical reactions by slowing molecular diffusion and reducing the encounter rates, thereby impeding diffusion-limited reactions. Conversely, the crowded environment enhances molecular assembly through entropic effects (the depletion attraction force) and promotes reaction-limited reactions. Recent findings have revealed that the cytoplasm is not merely a crowded environment but exhibits a complex porous architecture [105,106,107], which is not reproduced by in vitro assays such as bovine serum albumin (BSA) solutions [108,109]. Additionally, the cytoplasm is dynamically agitated by metabolic activities that arise from the combined effects of molecular motor motion, cytoskeletal structures and dynamics, and enzymatic reactions. Indeed, ATP depletion causes the cytoplasm of bacterial and yeast cells to exhibit solid or glass-like properties [103,110,111,112,113], highlighting the crucial role of ATP-driven intracellular processes in maintaining cytoplasmic fluidity. These properties, along with cytoplasmic viscoelasticity and pH levels, together contribute significantly to cellular organization and macromolecular movement [103,114].

Dormant cells exhibit more viscous, rigid, and dense cytoplasm, which is believed to contribute to their stress resistance. For example, the spore cytoplasm of *Bacillus subtilis* immobilizes proteins rotationally, potentially serving as a mechanism to prevent protein denaturation [115]. In more extreme cases, plant seeds and several animal species achieve immobilization of the entire cytoplasm through vitrification by replacing water with disaccharides [116,117,118]. However, some dormant cells exist in intermediate states, where their cytoplasm has low fluidity but still maintains metabolic activities. The cytoplasm of such intermediate dormancy states, including that of yeast spores, remains an active area of research.

#### 4.1.1. What Cellular Processes Are Affected by Cytoplasmic Properties?

The biophysical properties of the cytoplasm are modulated in response to environmental perturbations, including osmotic stress, temperature fluctuations, nutrient availability, and mechanical forces. These alterations in the cytoplasmic properties impact myriad biochemical reactions and cellular organization. A key example is microtubule dynamics, which are influenced by osmotic stress in fission yeast, mammalian, and moss cells [119]. Hyperosmotic stress reduces cell volume with water efflux, increasing cytoplasmic concentration, while hypoosmotic stress expands cell volume, decreasing the concentration. Microtubule polymerization and depolymerization rates vary linearly and inversely with cytoplasmic concentration, likely due to changes in cytoplasmic viscosity. Moreover, increased cytoplasmic crowding upon severe hyperosmotic stress slows down diffusion rates and nuclear import of stress-response signaling molecules in budding yeast cells [120]. Protein production and growth rate also decrease with increased cytoplasmic crowding upon osmotic stress, indicating that these processes are diffusion-limited [121]. Similarly, when yeast cells are grown in confined spaces, growth-induced mechanical pressure increases cytoplasmic crowding, which hinders gene expression and impedes growth rate [121]. Another important example involves liquid-liquid phase separation (LLPS), which is controlled by cytoplasmic crowding in yeast and mammalian cells [122]. When cells are treated with rapamycin, a mTORC1 inhibitor, ribosome concentration drastically decreases and particle diffusion increases, showing that ribosomes function as a cytoplasmic crowding agent. Phase separation using a synthetic system proportionally increases with ribosome concentration in the cytoplasm [122]. Finally, kinase phosphorylation of the ERK MAP kinase pathway is controlled by molecular crowding in mammalian cells [123]. The activation loop of ERK is processively phosphorylated by MEK kinase in cells. Processive phosphorylation can be reproduced in vitro with crowding agents, but without them, the reaction shifts to distributive phosphorylation. This suggests that cytoplasmic crowding plays a key role in regulating the successive phosphorylation reactions.

#### 4.1.2. Homeostasis Mechanisms Regulating Cytoplasmic Properties

Cells possess homeostasis mechanisms that control the biophysical properties of the cytoplasm and adapt to environmental changes. Recent studies have identified WNK1 as a molecular crowding sensor in mammalian cells, which facilitates cell volume recovery and alleviates crowding under hyperosmotic stress [124]. As another example, when exposed to elevated temperatures, budding yeast cells produce trehalose and glycogen to enhance cytoplasmic viscosity, thereby counteracting the increase in protein diffusivity [125]. Given that the cytoplasmic properties affect cellular reactions and processes, extreme changes in cytoplasmic biophysics can lead to cellular dysfunctions. Indeed, excessive cell growth without DNA replication causes cytoplasmic dilution, contributing to cellular senescence [126]. These homeostasis mechanisms are therefore critical for maintaining the proper cytoplasmic environment and cellular functions, though it is still unclear the causal relation between cytoplasmic properties and cellular functions.

#### 4.1.3. Methods to Evaluate the Cytoplasmic Properties

Various techniques allow us to measure the biophysical properties of the cytoplasm. Here, we focus on the cytoplasmic fluidity inside the cell, which can be measured by tracking the mobility of proteins, nucleic acids, and particles and computing their diffusion coefficients. These measurements enable us to estimate the viscoelasticity, molecular crowding, cellular structure, and organization inside the cell. The diffusion of molecules with an average protein size (~5 nm) can be assessed using several methods, such as fluorescence recovery after photobleaching (FRAP), fluorescence correlation spectroscopy (FCS), fluorescence loss in photobleaching (FLIP), and fluorescence decay after photostimulation (FDAP) [127]. Each technique offers unique benefits; for example, FRAP is well-suited for slow diffusion (<1.0 µm^2^/sec), while FCS specializes in fast diffusion (>1.0 µm^2^/sec). The mobility of large molecules (~10 to 1000 nm) can be assessed using particle-tracking microrheology [128]. Recently, genetically encoded nanoparticles have been developed and applied across various biological contexts [129]. These nanoparticles consist of a scaffolding domain and fluorescent protein, self-assembling into bright spherical particles inside the cell. Their size can be adjusted by substituting the scaffolding domain. Several types of nanoparticles are available, such as μNS (50–150 nm) [110] and genetically encoded multimeric nanoparticles (GEMs) with 20 nm (20 nm-GEMs), 40 nm (40 nm-GEMs), and 50 nm (50 nm-GEMs) in diameter [122,130]. In addition, GEMs can be localized to different organelles, including the nucleus (nucGEMs) [131] and endoplasmic reticulum (ER-AqLs) [132]. Genetically encoded nanoparticles overcome several technical challenges; for example, unlike endogenous structures, they are bio-orthogonal, and therefore they do not interact with intracellular components, allowing simplification of mobility analysis. Additionally, genetically encoded nanoparticles can be utilized in organisms with rigid cell walls, where microinjection of exogenous particles is not feasible. Genetically encoded nanoparticles have been successfully implemented in numerous biological systems, spanning bacteria, yeasts (*S. pombe*, *S. cerevisiae*, *C. albicans*), fungi (*Ashbya gossypii*), mammalian cells, and *Drosophila* [37,122,133,134,135]. This broad applicability highlights the versatility of this technique for studying cytoplasmic properties across evolutionarily distant organisms. Despite its utility, this methodology presents several important technical challenges. Notably, it exhibits high sensitivity to expression levels, potentially resulting in aggregate formation when overexpressed. Such aggregation may also be triggered under specific stress conditions, which can confound experimental interpretations and limit applicability in certain cellular states. Overall, genetically encoded nanoparticles have emerged as transformative tools for investigating cytoplasmic biophysics across numerous biological contexts. Their application over the past decade has catalyzed unprecedented advances in our understanding of intracellular biophysical properties, generating a wealth of evidence and insights that have fundamentally reshaped our conceptual framework in this field. A particularly exciting prospect for advancing this technique involves comprehensive genetic screening to identify key regulators of cytoplasmic properties. The methodology is ideally suited for organisms with rigid cell walls, such as bacteria and yeasts, which offer robust platforms for genetic approaches. Large-scale screens would likely offer unprecedented insights into the fundamental organization of the cytoplasm and potentially reveal novel mechanisms that orchestrate its biophysical regulation. This approach could bridge molecular genetics with biophysics, establishing new paradigms for understanding cellular dormancy and its reversal.

### 4.2. Dormancy and Dormancy Breaking

#### 4.2.1. The Biophysical Properties of the Cytoplasm in Dormant Yeast Spores

Two recent studies using microrheological measurements have demonstrated that the biophysical properties of the cytoplasm are drastically altered during dormancy in *S. pombe* and *S. cerevisiae* [37,62]. In *S. pombe*, the mobility of 40 nm-GEMs and μNS (50–150 nm) particles is strongly restricted in spores compared to vegetative cells [37]. Similarly, in *S. cerevisiae*, the mobility of μNS particles is reduced in spores [62], showing that spore cytoplasm hinders macromolecular diffusion during dormancy. In contrast to these particles, the fluorescent proteins, comparable to the average protein size (~5 nm), are relatively free to diffuse even in the spore cytoplasm in *S. pombe* [37]. This size-dependent limitation on cytoplasmic diffusion might explain how dormant cells, despite the diffusion and reaction barriers imposed by the crowded spore cytoplasm, can rapidly resume growth when environmental conditions improve. Specifically, it is proposed that small signaling molecules are relatively free to diffuse in the spore cytoplasm and transmit dormancy-breaking signals upon germination initiation, whereas the mobility of large protein complexes, such as ribosomes (~ 30 nm), is hindered. Recent in vitro studies with *E. coli* and *Xenopus* cytoplasms revealed that an overcrowded cytoplasm can inhibit ribosome mobility, leading to reduced translational activity [136,137]. Similarly, the crowded milieu of spore cytoplasm may suppress translational activity. Indeed, the protein expression rate slows down in spores compared to vegetative cells in *S. cerevisiae* [82]. The reduced protein production may help spores conserve energy sources (e.g., ATP) and substrates (e.g., amino acids, nucleic acids), thereby reducing essential metabolic activity and enabling their extended survival without nutrients.

#### 4.2.2. The Biophysical Properties of the Cytoplasm During Spore Germination

The biophysical properties of the cytoplasm during *S. pombe* germination are shown in Figure 4. Germination triggers dynamic changes in the spore cytoplasm. In *S. pombe*, 40 nm-GEMs mobility exhibits a rapid increase within an hour after germination induction [37]. This cytoplasmic fluidization may contribute to an increase in metabolic activity, driving the subsequent germination progression. Of note, the increased mobility of 40 nm-GEMs precedes a bright-to-dark transition under phase contrast microscopy, the traditionally recognized earliest hallmark of germination initiation [37,47,48]. In both *S. pombe* and *S. cerevisiae*, larger particles (μNS, 50–150 nm) behave slightly differently from 40 nm-GEMs, showing gradually increasing mobility during germination [37,62]. These results suggest that cellular structures are reorganized on a scale of 50 to 150 nm over time. *S. cerevisiae* spores contain cytoplasmic foci of specific proteins (e.g., Acc1, Ura7), which gradually disassemble and then diffuse throughout the cytoplasm as μNS mobility increases [62]. The overcrowded spore cytoplasm might entropically drive these cytoplasmic foci, while cytoplasmic fluidization may accelerate their disassembly. This proposed mechanism is supported by the finding that Acc1 foci formation is facilitated by crowding agents [138].

#### 4.2.3. The Mechanisms Regulating Cytoplasmic Properties During Dormancy and Germination

Previous studies have proposed several mechanisms for the molecular and physicochemical regulation of the cytoplasmic properties in vegetative yeast cells and other organisms: (1) pH levels [103], (2) cellular volume changes [111,119], (3) cytoskeleton structure and dynamics [139], (4) viscogen concentration (e.g., trehalose, glycogen, sorbitol) [125,140], (5) ATP-driven metabolic activity [110], and (6) cytoplasmic crowder concentration (e.g., polysome, ribosome, and RNA) [112,122,141]. Some of these potential mechanisms have been investigated during dormancy and germination in *S. pombe* and *S. cerevisiae* spores. First, pH-driven cytoplasmic reorganization is known to trigger cytoplasmic solidification. In *S. cerevisiae*, the pH level in dormant spores (pH ~5.9) drops and then increases during germination, eventually approaching the value found in vegetative cells (pH ~7.4) [62,63]. However, the increase in μNS mobility is delayed compared to an increase in intracellular pH level. These findings suggest that while decreased pH levels may contribute to cytoplasmic properties during dormancy, other mechanisms must also be involved in the regulation of the cytoplasmic fluidity. In *S. pombe*, while spore germination involves cell swelling, the cytoplasmic fluidization based on 40 nm-GEMs mobility precedes the changes in cell sizes [37]. Thus, water uptake accompanied by cell swelling may not be required for cytoplasmic fluidization during germination. Furthermore, actin and microtubule dynamics are not involved in cytoplasmic fluidization in *S. pombe* germinating spores [37]. A recent study has proposed that trehalose, known to act as cytoplasmic viscogen [125], plays a key role in regulating cytoplasmic fluidity during dormancy and germination in *S. pombe* [37]. *S. pombe* spores contain large amounts of trehalose during dormancy [37,38]. Upon germination initiation, the trehalose is quickly degraded concurrently with increased 40 nm-GEM mobility. This trehalose degradation requires the glucose-sensing by cAMP-PKA pathway and natural trehalase Ntp1. Deletion of genes encoding components in cAMP-PKA-Ntp1 signaling impedes trehalose degradation and cytoplasmic fluidization, suggesting a direct link between trehalose and cytoplasmic fluidity. Similarly, in *S. cerevisiae*, trehalose accumulates in spores and rapidly degrades during germination [62], suggesting a conserved mechanism for controlling cytoplasmic properties. While metabolic activity is generally reduced during dormancy, ATP-driven processes could influence cytoplasmic properties [37]. Further investigation is also required to verify whether cytoplasmic crowder concentration regulates cytoplasmic properties during dormancy and germination in yeast cells.

## 5. Summary and Perspective

In this review, we have explored the molecular and biophysical aspects of dormancy and dormancy breaking in yeast models. Numerous studies using *S. cerevisiae* and *S. pombe* have elucidated the complex molecular machinery governing these processes, including signaling pathways, metabolic adaptations, and gene regulatory networks. Recent findings have particularly highlighted the crucial role of cytoplasmic biophysical properties in dormancy and its breaking. Several key questions remain to be addressed: (1) How do cytoplasmic properties regulate cellular functions during dormancy and their reactivation during germination? In particular, it remains unclear whether dormancy-specific proteins undergo phase separation, reminiscent of the reversible phase separation observed for the prion-like protein FLOE1 during seed germination [142]. (2) What is the mechanistic link between trehalose accumulation and cytoplasmic properties in dormant cells? (3) How does dormancy affect the diffusion of endogenous proteins? (4) What are the biophysical properties of other cellular components, such as the nucleus, during dormancy? In addition to the biophysical aspects, critical questions remain in other areas: (1) It is still technically challenging to identify genes required specifically for germination but not for meiosis and sporulation. Future improvements in genome-wide screening would offer new insights into the molecular mechanisms during germination. (2) There is limited knowledge regarding the advantages of sporulation in natural environments, particularly in *S. pombe*. *S. pombe* in its natural habitats might possess unique sporulation characteristics not observed in laboratory strains. In the case of *S. japonicus*, a species in the same genus as *S. pombe*, the naturally isolated strains from *Drosophila* sporulate more efficiently than wild strains isolated from fruits and plants [143].

The comparative study of dormancy mechanisms across different yeast species offers valuable evolutionary insights. Similarities and differences in how these distantly related yeasts regulate dormancy may reveal core conserved mechanisms essential for cellular survival as well as species-specific adaptations to particular ecological niches. Advanced techniques such as cryo-electron tomography, super-resolution microscopy, and single-molecule tracking will likely provide unprecedented views of the dormant cellular state at the molecular level in coming years.

## 6. Conclusions

In this review, we have provided a comprehensive overview of dormancy and germination in two model yeast species, *S. pombe* and *S. cerevisiae*, with particular emphasis on molecular mechanisms and biophysical properties. Using these simple model systems, researchers have elucidated comprehensive changes in gene and protein expression, metabolic profiles, as well as alterations in the biophysical properties of the cytoplasm and its control during dormancy and germination. The cytoplasm in dormant spores differs markedly from that in vegetative cells, characterized by an acidic, viscous, and overcrowded environment. The integration of molecular, biophysical, and ecological approaches in yeast dormancy research continues to reveal fundamental principles of cellular survival. These insights not only deepen our understanding of basic cell biology but also have important implications for biotechnology and medicine, where controlling cellular dormancy is of crucial importance. Future studies focusing on the mechanistic links between cytoplasmic properties and cellular functions during dormancy, as well as improved genome-wide screening methods to identify germination-specific genes, will further advance our understanding of these remarkable survival strategies that are conserved across diverse species.

## Figures and Tables

**Figure 1 biomolecules-15-00701-f001:**
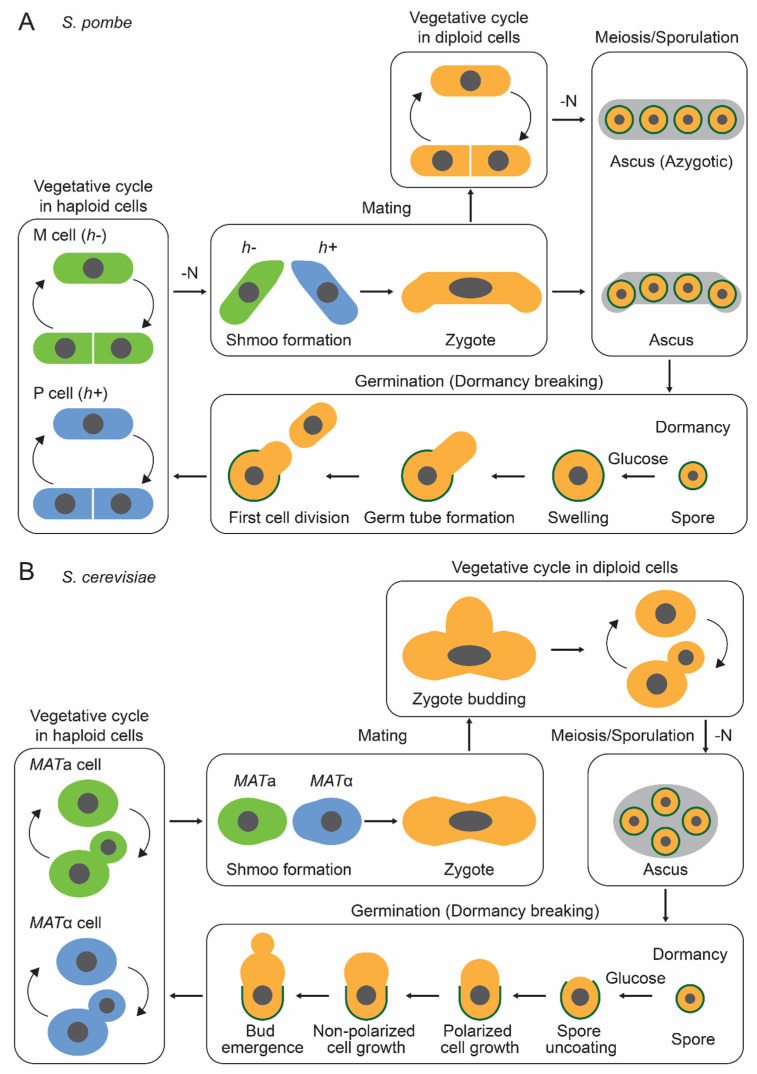
Life cycle for *S. pombe* and *S. cerevisiae*. Haploid yeast cells proliferate under nutrient-rich conditions. In *S. pombe* (**A**), nitrogen starvation (N) induces the mating of opposite mating types (h− and h+), followed by meiosis and sporulation, resulting in ascus formation. If zygotes receive nutrients before progressing through these processes, they can proliferate as diploid cells. These diploid cells undergo meiosis and sporulation in response to nitrogen starvation, forming an azygotic ascus, which has a distinct morphology from the normal ascus. These azygotic ascus and zygotic ascus naturally rupture, releasing dormant spores. When spores sense the presence of glucose, they initiate the germination process (dormancy breaking), followed by spore swelling and germ tube formation (outgrowth). After the first cell division, they re-enter the vegetative cycle. In *S. cerevisiae* (**B**), even under nutrient-rich conditions, cells of opposite mating types (*MATa* and *MATα*) can mate and proliferate as diploid cells. In response to nitrogen starvation, diploid cells undergo meiosis and sporulation, forming a zygotic ascus. The ascus remains intact for an extended period, unlike *S. pombe*. During germination, spores undergo several morphological changes, such as spore uncoating, polarized cell growth, non-polarized cell growth, and bud emergence.

**Figure 2 biomolecules-15-00701-f002:**
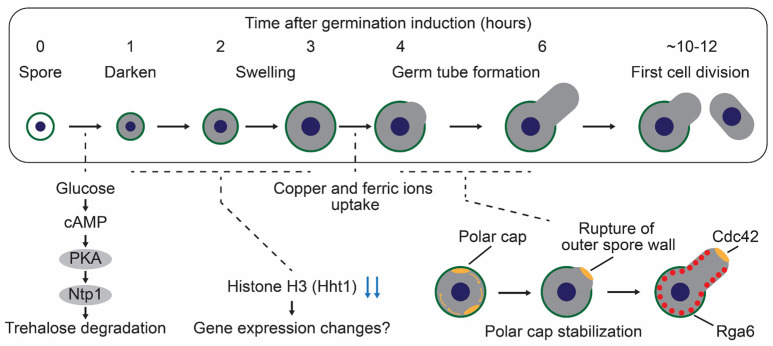
The events during spore germination in *S. pombe*. First, fission yeast spores sense and respond to glucose through the cAMP-PKA signaling pathway, which leads to Ntp1-mediated trehalose degradation. Subsequently, the expression level of histone H3 Hht1 is decreased, potentially facilitating chromatin relaxation and inducing global changes in gene expression. Additionally, the uptake of copper and ferric ions is indispensable for germ tube formation (outgrowth). In swollen spores, the polar cap, including a Rho-type GTPase Cdc42, moves dynamically along the cell surface before stabilizing at a specific site, where the outer spore wall ruptures to initiate germ tube formation (orange dots). This monopolar outgrowth requires Rga6, a Cdc42 GAP (red dots), which determines the tip localization of Cdc42.

**Figure 3 biomolecules-15-00701-f003:**
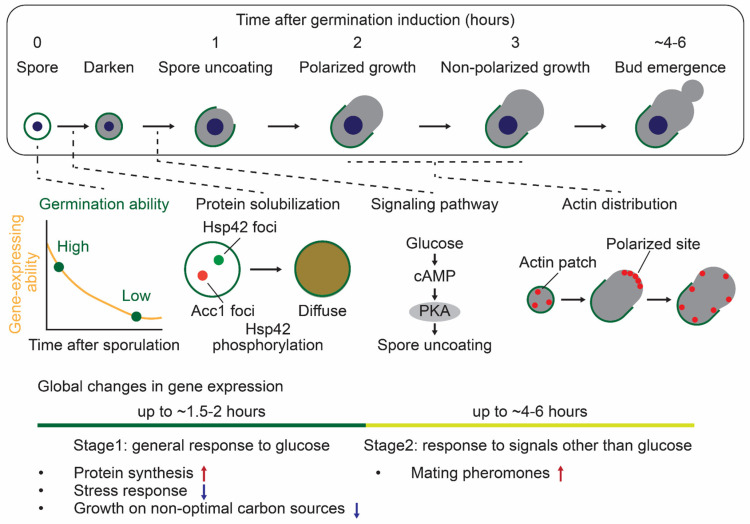
The events during spore germination in *S. cerevisiae*. After the completion of sporulation, the germination ability of budding yeast spores gradually decreases over time. This decline is due to the decreased gene-expressing ability, such as a decrease in the RNA polymerase II amount. In the presence of glucose, spores sense and respond to it through the cAMP-PKA signaling pathway, initiating germination. In the early stage of germination, small heat shock protein Hsp42 foci dissolved upon its phosphorylation, promoting the solubilization of other proteins, including the metabolic enzyme Acc1. Subsequently, actin patches (red dots) accumulate at the tip region to facilitate polarized growth and later disperse again, potentially promoting non-polarized cell growth. Furthermore, gene expression undergoes global changes following germination induction. The gene expression pattern can be categorized into two stages: the general response to glucose (up to ~1.5–2 h) and the response to signals other than glucose (up to ~4–6 h).

**Figure 4 biomolecules-15-00701-f004:**
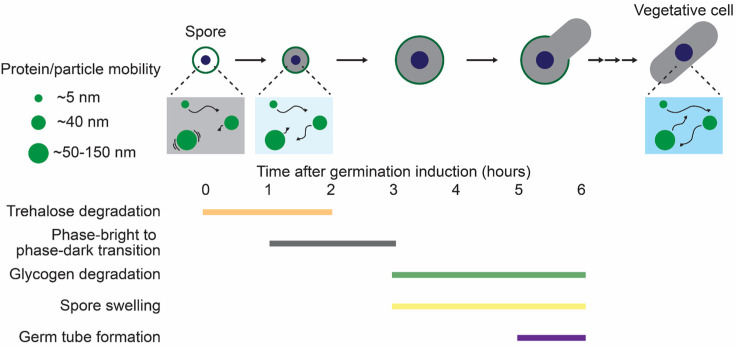
The biophysical properties of the cytoplasm during dormancy and germination in *S. pombe*. Protein diffusion and particle mobility are illustrated with the timing of the germination events. In dormant spores, the mobility of particles (~40 nm and ~50–150 nm in diameter) is restricted, whereas small proteins can diffuse relatively freely. Upon germination induction, particle mobility increases rapidly.

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
