# Peer review of "Molecular and Biophysical Perspectives on Dormancy Breaking: Lessons from Yeast Spore"

_biomolecules, 2025, doi:10.3390/biom15050701_

Round 1
Reviewer 1 Report
Comments and Suggestions for Authors
Overall, the review was well-written and easy to read.
Positives:
The comparison and contrast of S. cerevisiae and S. pombe was informative.
Overall, the writing was excellent and it serves as a nice reference on the topic.
Potential ways to revise/improve:
- There was some redundancy relating mostly to section 2 (mostly older literature) and lack of depth on others (sections 3 and 4; mostly newer literature). Neither of these critiques are meant to be strongly negatives, rather they are merely mentioned as a possible way to improve the review if a revision is desirable.
- The review was a little dry – the authors covered literature well, but it would have been interesting for the authors to speculate a bit more about what they see as the cutting edge of this research, adding their own expert perspectives, and to reduce the (sometimes redundant) discussion of dogma that they agree with and delve a little deeper into newer aspects of the topic. This could include providing more in depth discussion and elaboration on the summary of the most cutting edge references, and also highlighting any controversial areas of the research, assuming they exist, and how best to resolve them.
- If possible, it would be interesting for the authors to expand upon dormancy breaking, single-cell analyses, and biophysical studies. The suggestion is not necessarily to add more references, but rather to review the cited literature in greater depth and infusing more of their opinion and interpretation about the strongest, most compelling, and most promising directions for the field moving forward.
- Expansion (strengths, weaknesses, how transformative they are for field, future directions) on the methods to evaluate cytoplasmic properties (Section 4.1.3) would probably be an improvement.
- One consideration would be to organize the review as outlined in the Summary and Perspective.
Author Response
Comments 1:There was some redundancy relating mostly to section 2 (mostly older literature) and lack of depth on others (sections 3 and 4; mostly newer literature). Neither of these critiques are meant to be strongly negatives, rather they are merely mentioned as a possible way to improve the review if a revision is desirable.
Response 1: We appreciate this reviewer’s comment. We understand the concern that some parts of Section 2 may appear redundant, possibly due to the separate discussion of S. pombe and S. cerevisiae. However, this deliberate organizational choice is a distinctive strength of our review. Most previous reviews on sporulation and germination typically either focus on just one yeast species or merge findings from both species within the same sections. By maintaining separate sections for each species, we enable readers to understand the unique aspects of each organism better while still facilitating cross-species comparisons. After careful consideration, we have maintained this structure, as we believe it provides significant value to researchers interested in either or both model systems.
We have added the additional discussions related to Sections 3 and 4. Please refer to our responses to Comments 3 and 4.
Comments 2: The review was a little dry – the authors covered literature well, but it would have been interesting for the authors to speculate a bit more about what they see as the cutting edge of this research, adding their own expert perspectives, and to reduce the (sometimes redundant) discussion of dogma that they agree with and delve a little deeper into newer aspects of the topic. This could include providing more in depth discussion and elaboration on the summary of the most cutting edge references, and also highlighting any controversial areas of the research, assuming they exist, and how best to resolve them.
Response 2: Thank you for your thoughtful, constructive feedback. As noted in our responses to Comments 3 and 4, we have substantially expanded our discussion to incorporate insights from recent cutting-edge research.
Comments 3: If possible, it would be interesting for the authors to expand upon dormancy breaking, single-cell analyses, and biophysical studies. The suggestion is not necessarily to add more references, but rather to review the cited literature in greater depth and infusing more of their opinion and interpretation about the strongest, most compelling, and most promising directions for the field moving forward.
Response 3: Thank you for your constructive suggestions. We have substantially enhanced our manuscript with additional explanations related to the promising future directions in dormancy breaking, single-cell analysis, and biophysical studies as follows.
Dormancy breaking
(Section 3.1.1.) 
“In the future, it will be of particular interest to identify downstream factors of the cAMP-PKA pathway and to elucidate their specific roles during germination. In S. pombe, a comprehensive analysis of PKA substrates remains unavailable, even for vegetative growth conditions. Consequently, our understanding of molecular pathways activated downstream of glucose-induced PKA signaling is limited, with only a few identified targets such as Ntp1. A promising approach would involve integrated transcriptome, proteome, and phosphoproteome analysis of the pka1Δ strain in vegetative cells, as synchronizing germination in S. pombe spores presents significant technical challenges (see section 3.1.2. Gene expression landscape during germination in S. pombe). Such multi-omics would likely reveal the comprehensive molecular networks that orchestrate the initiation of germination.”
(Section 3.1.3.)
“In the future, a systematic investigation of nutrient requirements for germination would yield valuable insights into the molecular mechanisms that drive germination progression. Currently, glucose function is well established; it acts both as a primary carbon source and an activator of the cAMP-PKA signaling pathway. In contrast, our knowledge of other nutrients, particularly copper and iron ions, remains limited, and many potentially essential nutrients await discovery and characterization. Deciphering the complete nutritional landscape necessary for germination would likely reveal important regulatory mechanisms governing this fundamental biological process.”
(Section 3.2.4.)
“Investigating the relationship between enzyme solubilization and subsequent enzymatic activity during the initiation of germination represents a compelling avenue for future research.”
Single-cell analysis
(Section 3.1.2.)
“Determining the precise functions of these candidate genes during germination would be a valuable future direction. In addition, the established single-cell transcriptome methodology offers powerful opportunities when applied to mutant strains for characterizing gene expression dynamics throughout germination. A promising candidate for such analysis is Pka1, as discussed in Section 3.1.1, which could illuminate the molecular mechanisms governing germination in S. pombe.”
Biophysical studies
(Section 4.1.3.)
Please refer to the responses to Comment 4.
In addition, about biophysical studies, we discussed the promising future directions and questions in Section 5. Summary and perspective.
Comments 4: Expansion (strengths, weaknesses, how transformative they are for field, future directions) on the methods to evaluate cytoplasmic properties (Section 4.1.3) would probably be an improvement.
Response 4: We appreciate this insightful suggestion and have substantially expanded Section 4.1.3 to provide a more comprehensive assessment of methods for evaluating cytoplasmic properties. As the reviewer recommended, we have enhanced this section with the following parts:
Strengths
“Genetically encoded nanoparticles have been successfully implemented in numerous biological systems, spanning bacteria, yeasts (S. pombe, S. cerevisiae, C. albicans), fungi (Ashbya gossypii), mammalian cells, and Drosophila. This broad applicability highlights the versatility of this technique or studying cytoplasmic properties across evolutionarily distant organisms.”
Weaknesses
“Despite its utility, this methodology presents several important technical challenges. Notably, it exhibits high sensitivity to expression levels, potentially resulting in aggregate formation when overexpressed. Such aggregation may also be triggered under specific stress conditions, which can confound experimental interpretations and limit applicability in certain cellular states.”
How transformative they are for the field
“Overall, genetically encoded nanoparticles have emerged as transformative tools for investigating cytoplasmic biophysics across numerous biological contexts. Their application over the past decade has catalyzed unprecedented advances in our understanding of intracellular biophysical properties, generating a wealth of evidence and insights that have fundamentally reshaped our conceptual framework in this field.”
Future directions
“A particularly exciting prospect for advancing this technique involves comprehensive genetic screening to identify key regulators of cytoplasmic properties. The methodology is ideally suited for organisms with rigid cell walls, such as bacteria and yeasts, which offer robust platforms for genetic approaches. Large-scale screens would likely offer unprecedented insights into the fundamental organization of the cytoplasm and potentially reveal novel mechanisms that orchestrate its biophysical regulation. This approach could bridge molecular genetics with biophysics, establishing new paradigms for understanding cellular dormancy and its reversal.”
Comments 5: One consideration would be to organize the review as outlined in the Summary and Perspective.
Response 5: We appreciate this thoughtful suggestion regarding the organizational structure of our manuscript. After careful consideration, we have decided to maintain our current organization with separate sections for S. pombe and S. cerevisiae. This approach represents a deliberate and distinctive feature of our review that differentiates it from existing literature in the field.
The majority of previous reviews on sporulation and germination have either focused exclusively on a single yeast species or merged findings from multiple species within unified sections. While such approaches have their merits, we believe our species-specific organization offers several unique advantages:
- It facilitates direct species-to-species comparisons of homologous processes
- It highlights species-specific adaptations and evolutionary divergence
- It provides clarity for researchers primarily interested in one model organism
- It avoids oversimplification when mechanisms differ significantly between species
Given that these two yeast species diverged approximately 330-420 million years ago, their distinct evolutionary trajectories have resulted in both conserved and divergent mechanisms governing dormancy and germination. Our current structure preserves these important nuances while still enabling synthetic insights in the Summary and Perspective sections.
We believe this organizational approach delivers maximum value to our readers, particularly those conducting comparative analyses or seeking a comprehensive understanding of either model system.
Reviewer 2 Report
Comments and Suggestions for Authors
This review focusses on the sporulation process, spore properties and germination in both S. pombe and S. cerevisiae.
Major comments
1- By contrast to what is stated line 84 "We also explore how insights from yeast studies contribute to our understanding of cellular dormancy across species", in this review, no significant comparison with other species are provided. Therefore, the authors may consider to soften this statement and change the title of the review.
2- It may help the non specialist reader to clearly state the definition of quiescence, dormancy, cryptobiosis... and compare these states. Another option could be to simply remove all the references to quiescence (lines 75, 76, 77, line 127).
3- It may be interesting to distinguish yeast spore dormancy and dormancy in plants.
4- The autors compare S. pombe and S. cerevisiae and such, it would be interesting to comment on the phylogenetic distance between the two species.
Minor comments
1- Line 70 it is stated that "Nutrient limitation induces the formation of spores, a dormant state, in these yeasts [9]." This is not really acurate. In fact NITROGEN starvation causes sporulation of DIPLOID yeast.
2- Line 183 The authors stated that "Using doxycycline-inducible synthetic gene-circuit, it was demonstrated that GFP expression in spores requires approximately 20 hours to reach plateau levels, compared to 8 hours in vegetative cells, revealing that gene expression slows down in spores [45]."
GFP needs oxygen to fold and glow. It may be that GFP folding rather than gene expression is slowed down in spores....
3- The ATP involvement need to be clarified. On the one hand, it is stated line 188 that “pharmacological inhibition of ATP synthesis does not compromise spore viability [45]”. On the other hand, line 196, it is suggested that cells should "maintain ATP levels necessary for minimal metabolic activity [46]. Indeed, metabolomic analyses have revealed that spores maintain substantial amounts of ATP (~3–4 mM) [47]”. So it is not clear if spores do have to maintain their ATP level or if is not necessary.
4-The paragraph "3.1.2. Gene expression landscape during germination in S. pombe" is not really discussing gene expression but rather the function of histones.
5-The paragraph 4.1.3. Methods to evaluate cytoplasmic properties may be shortened as not all information is relevant.
Comments on the Quality of English Language
The quality of the English language should be significantly improved.
Some sentences are really awkward.
For exemple line 68 "Yeasts, particularly two big model yeasts Saccharomyces cerevisiae and Schizosaccharomyces pombe, have emerged as powerful model systems for studying cellular dormancy." --> should be changed for exemple by "Saccharomyces cerevisiae and Schizosaccharomyces pombe have emerged has powerfull models to study cellular dormancy".
Line 225 the autors have writen : "the ascus walls are digested, causing the dissolution of each spore" do they mean "dissemination of each spore" ?
Author Response
Major comments 1: By contrast to what is stated line 84 "We also explore how insights from yeast studies contribute to our understanding of cellular dormancy across species", in this review, no significant comparison with other species are provided. Therefore, the authors may consider to soften this statement and change the title of the review.
Response 1: We appreciate the reviewer's careful reading and thoughtful feedback regarding our statement on line 84. We agree that our manuscript does not provide extensive cross-species comparisons beyond the two yeast model systems. Accordingly, we have changed this statement to more accurately reflect the scope of our review as below.
“We further explore emerging questions in this rapidly evolving field, with particular focus on molecular, biophysical, and ecological perspectives.”
This modification better aligns with our actual content and research focus. Regarding the title, we have chosen to retain it because it precisely delimits our subject matter without implying broader cross-species analysis. The title accurately communicates to readers that our review comprehensively examines dormancy and germination specifically within yeast models, which represent foundational systems for understanding these processes. We believe this title appropriately positions our work within the literature while maintaining clarity about its scope and contribution to the field.
Major comments 2: It may help the non specialist reader to clearly state the definition of quiescence, dormancy, cryptobiosis... and compare these states. Another option could be to simply remove all the references to quiescence (lines 75, 76, 77, line 127).
Response 2: Thank you for the constructive suggestions to improve clarity for non-specialist readers. Rather than removing references to quiescence, we have opted to add clear definitions that distinguish between these related physiological states. As noted in a recent review (Rachel, 2023), these terms are often loosely defined in the literature, but important distinctions exist. We have added the following clarification here:
“Both dormancy and quiescence refer to reversible growth-arrested states, but dormancy is generally thought to be a “deeper” form of quiescence (Rachel, 2023). Quiescent cells typically retain higher levels of intracellular activity compared to dormant cells.”
Additionally, we have added the explanations for related terms, cryptobiosis and anhydrobiosis, as follows.
“cryptobiosis (meaning ‘hidden life’), a term coined by David Keilin (Proc. Roy. Soc. Lond. B, 150, 1959, 149-191) and defined as 'the state of an organism when it shows no visible signs of life and when its metabolic activity becomes hardly measurable, or comes reversibly to a standstill.”
“anhydrobiosis (meaning ‘life without water’), a state of cryptobiosis that is induced by desiccation”
We believe these additions significantly enhance the accessibility of our review for readers across disciplines without disrupting the overall flow of the manuscript.
Major comments 3: It may be interesting to distinguish yeast spore dormancy and dormancy in plants.
Response 3: Thank you for this thoughtful suggestion regarding the comparison between yeast and plant dormancy. While we recognize the potential value of such a comparison, we have made a strategic decision to maintain our review's focused scope on yeast systems. Expanding into detailed discussions of plant dormancy mechanisms would significantly broaden the manuscript and potentially dilute our in-depth analysis of yeast-specific processes. Plant dormancy encompasses complex systems, including seed dormancy, bud dormancy, and seasonal vegetative dormancy, each with distinct regulatory mechanisms that would require extensive treatment to address properly.
Instead, we have taken a balanced approach by briefly contextualizing yeast dormancy within the broader biological phenomenon in Section 1 (Introduction). Additionally, in Section 5 (Summary and perspective), we discuss recent significant findings in plant dormancy research that provide interesting comparative insights without shifting our primary focus.
Major comments 4: The autors compare S. pombe and S. cerevisiae and such, it would be interesting to comment on the phylogenetic distance between the two species.
Response 4: Thank you for this insightful suggestion regarding evolutionary context. We have added the following explanation regarding the phylogenetic relationship between S. cerevisiae and S. pombe:
“These two yeast species diverged approximately 330–420 million years ago, with an evolutionary distance comparable to that between yeasts and mammals (Sipiczki, 2000; Wood et al., 2002). Accordingly, comparative studies of these phylogenetically distant yeasts are expected to provide insights into both the evolutionary conservation and diversification of cellular dormancy.”
This evolutionary perspective strengthens our comparative approach by highlighting that S. cerevisiae and S. pombe represent distinct lineages that have evolved independently for hundreds of millions of years.
Minor comments 1: Line 70 it is stated that "Nutrient limitation induces the formation of spores, a dormant state, in these yeasts [9]." This is not really acurate. In fact NITROGEN starvation causes sporulation of DIPLOID yeast.
Response 5: We appreciate the reviewer for this important correction. We have revised this sentence to accurately reflect the specific conditions required for sporulation.
“Nitrogen starvation induces the formation of spores, a dormant state, in these yeasts when they exist as diploid cells.”
Minor comments 2: Line 183 The authors stated that "Using doxycycline-inducible synthetic gene-circuit, it was demonstrated that GFP expression in spores requires approximately 20 hours to reach plateau levels, compared to 8 hours in vegetative cells, revealing that gene expression slows down in spores [45]."
GFP needs oxygen to fold and glow. It may be that GFP folding rather than gene expression is slowed down in spores....
Response 6: We agree with this reviewer’s comment. This comment highlights an important methodological consideration that we had not adequately addressed in our original manuscript. We made the following changes in this sentence.
“Using a doxycycline-inducible synthetic gene-circuit, it was demonstrated that GFP expression reaches a plateau after approximately 20 hours in spores, compared to 8 hours in vegetative cells. This observation suggests a slowdown in gene expression in spores, although the delay may also reflect the unique physiological environment within spores that affects oxygen-dependent GFP chromophore maturation.”
Minor comments 3: The ATP involvement need to be clarified. On the one hand, it is stated line 188 that “pharmacological inhibition of ATP synthesis does not compromise spore viability [45]”. On the other hand, line 196, it is suggested that cells should "maintain ATP levels necessary for minimal metabolic activity [46]. Indeed, metabolomic analyses have revealed that spores maintain substantial amounts of ATP (~3–4 mM) [47]”. So it is not clear if spores do have to maintain their ATP level or if is not necessary.
Response 7: We appreciate your suggestion highlighting this important conceptual issue regarding ATP metabolism in dormant spores. We agree that our original text contained an apparent contradiction that required clarification. We have expanded this section with the following explanations.
“Since transcription initiation requires ATP as an energy source (Michele, JBC, 1983), it is speculated that dormant spores retain sufficient ATP to sustain minimal transcription for at least 24 hours in the absence of ATP synthesis.”
Minor comments 4: The paragraph “3.1.2. Gene expression landscape during germination in S. pombe” is not really discussing gene expression but rather the function of histones.
Response 8: Thank you for your valuable comments regarding Section 3.1.2. We agree that our original discussion placed disproportionate emphasis on histone function rather than providing a comprehensive overview of the gene expression landscape during germination in S. pombe.
To address this imbalance, we have added the following description, including the broader characterization of gene expression landscape during germination.
“In this article, the authors identified 167 genes exhibiting dynamic expression changes during the early stage of germination. The enrichment analysis revealed associations with mRNA metabolic process, lipid metabolic process, nucleotide-containing small molecule metabolic process, ascospore formation, and detoxification. Among these, hht1, encoding histone H3, emerged as a particularly intriguing candidate.”
Minor comments 5: The paragraph 4.1.3. Methods to evaluate cytoplasmic properties may be shortened as not all information is relevant.
Response 9: Thank you for this suggestion regarding Section 4.1.3. After careful consideration, we have decided to maintain the comprehensive coverage of methodological approaches in this section, as they provide essential context for our subsequent analyses of cytoplasmic properties during dormancy and germination.
This paragraph primarily introduces two methodological approaches. The first involves techniques for measuring the diffusion of molecules of average protein size (~5 nm), such as FRAP and FCS, which are relevant to our discussion in paragraph 4.2.1 regarding the diffusion behavior of fluorescent proteins (~5 nm) in spores. These methodological foundations are necessary for readers to properly interpret our findings on diffusional constraints in dormant states.
The second approach covers genetically encoded nanoparticles, including GEM and μNS, which are used to assess particle diffusion both during dormancy and germination. These tools have proven particularly valuable for characterizing the dramatic biophysical transitions that accompany dormancy entry and exit, revealing properties that cannot be detected using conventional techniques.
Rather than viewing these methodological details as extraneous, we consider them integral to the conceptual framework of our review, providing readers, particularly those new to the field, with the necessary background to critically evaluate current research and design future investigations. The techniques described have been instrumental in recent breakthroughs in understanding cytoplasmic phase transitions during dormancy, which we discuss extensively in later sections.
Comments on the Quality of English Language
The quality of the English language should be significantly improved.
Some sentences are really awkward.
For exemple line 68 "Yeasts, particularly two big model yeasts Saccharomyces cerevisiae and Schizosaccharomyces pombe, have emerged as powerful model systems for studying cellular dormancy." --> should be changed for exemple by "Saccharomyces cerevisiae and Schizosaccharomyces pombe have emerged has powerfull models to study cellular dormancy".
Response 10: Thank you for your constructive feedback regarding the quality of English in our manuscript. We acknowledge that some sentences could be improved for clarity and conciseness. Following your suggestion, we have modified line 68 as recommended.
“Saccharomyces cerevisiae and Schizosaccharomyces pombe have emerged as powerful models for studying cellular dormancy.”
Line 225 the autors have writen : “the ascus walls are digested, causing the dissolution of each spore” do they mean “dissemination of each spore” ?
As the reviewer pointed out, we meant “dissemination”, not “dissolution”. We have made the following changes.
“.....the ascus walls are digested, causing the dissemination of each spore [64]. This dissemination of spores enhances outbreeding rates….”
In response to these and other reviewers' comments, we have conducted a thorough review of the entire manuscript to improve clarity and readability. We have addressed all specific instances pointed out by reviewers and have made numerous additional improvements throughout the text. These revisions have substantially enhanced the overall language quality, ensuring precise scientific communication while maintaining appropriate academic tone. The manuscript will also undergo basic English editing through the publisher's service, further refining the language.
We believe these combined efforts have significantly improved the manuscript's readability and clarity. However, we remain receptive to addressing any additional passages that may still hinder comprehension of the scientific content.
Reviewer 3 Report
Comments and Suggestions for Authors
Sakai, et al. present a well-crafted review of the molecular mechanisms underlying sporulation and germination in yeasts. The discussion of the biophysical changes that occur in the cytoplasm of spores is likely to be of particular interest to the field. The manuscript is clearly written and includes very nice diagrams illustrating processes underlying sporulation, etc. I have only a few minor points:
- Table 1 contains a lot of useful information that is difficult to read due to its being spread out over several pages. I recommend the authors convert it to landscape format in order to get it to fit on a single page.
- The reference, Rittershaus, et al., is cited twice.
- "Bacillus subtilis" in line 46 should be italicized.
- I recommend removing the word "turning" in line 138, or clarifying what is meant by this.
- In line 394, "closely" should be changed to "close."
- I suggest changing "tend to germinate more frequently" in line 449 to "are more likely to germinate," as this is more precise considering a spore can only germinate once.
- "Dynamical" in line 483 should be changed to "dynamic."
Author Response
Comments 1: Table 1 contains a lot of useful information that is difficult to read due to its being spread out over several pages. I recommend the authors convert it to landscape format in order to get it to fit on a single page.
Response 1: Thank you for highlighting this important presentation issue that will enhance the accessibility of our review for readers. We appreciate your concern about the readability of this information-rich table.
The current format of Table 1 adheres to the journal's submission template requirements. However, we fully agree that a landscape orientation would significantly improve readability and allow readers to view the complete table on a single page.
While the final formatting decisions ultimately rest with the publisher's production team, we will explicitly request landscape orientation for Table 1 during the proof review stage.
Comments 2: The reference, Rittershaus, et al., is cited twice.
Response 2: Thank you for pointing out corrections. We have corrected this mistake.
Comments 3: “Bacillus subtilis” in line 46 should be italicized.
Response 3: We have corrected this mistake.
Comments 4: I recommend removing the word "turning" in line 138, or clarifying what is meant by this.
Response 4: We have removed the word “turning”.
Comments 5: In line 394, "closely" should be changed to "close."
Response 5: We have corrected this mistake.
Comments 6: I suggest changing "tend to germinate more frequently" in line 449 to "are more likely to germinate," as this is more precise considering a spore can only germinate once.
Response 6: We have changed as suggested.
Comments 7: "Dynamical" in line 483 should be changed to "dynamic."
Response 7: We have changed to “dynamic”.